# PARP-1 Expression and *BRCA1* Mutations in Breast Cancer Patients’ CTCs

**DOI:** 10.3390/cancers14071731

**Published:** 2022-03-29

**Authors:** Thodoris Sklias, Vasileios Vardas, Evangelia Pantazaka, Athina Christopoulou, Vassilis Georgoulias, Athanasios Kotsakis, Yiannis Vasilopoulos, Galatea Kallergi

**Affiliations:** 1Laboratory of Genetics, Section of Genetics, Cell Biology and Development, Department of Biology, University of Patras, 26504 Patras, Greece; thodorissklias@gmail.com; 2Laboratory of Biochemistry/Metastatic Signaling, Section of Genetics, Cell Biology and Development, Department of Biology, University of Patras, 26504 Patras, Greece; up1088956@upatras.gr (V.V.); evapantazaka@upatras.gr (E.P.); 3Oncology Unit, ST Andrews General Hospital of Patras, 26332 Patras, Greece; athinachristo@hotmail.com; 4Hellenic Oncology Research Group (HORG), 11526 Athens, Greece; georgulv@otenet.gr; 5Department of Medical Oncology, University General Hospital of Larisa, 41334 Larisa, Greece; thankotsakis@hotmail.com

**Keywords:** PARP-1, *BRCA1*, circulating tumor cells, breast cancer, triple negative breast cancer, luminal

## Abstract

**Simple Summary:**

Recent estimates have shown that approx. 70% of individuals with *BRCA1* mutations will develop breast cancer by the age of 70. To make matters worse, breast cancer patients with *BRCA1* mutations are more likely to have the more aggressive triple-negative breast cancer. PARPs, belong to a family of nuclear enzymes, which are involved in many cellular processes, including DNA repair. PARP inhibitors have been approved for the treatment of BRCA-mutated breast cancer. The aim of the study was the determination of PARP-1 expression in the context of the presence of *BRCA1* mutations in circulating tumor cells of breast cancer patients. PARP-1 (nuclear) expression and *BRCA1* mutations were mainly detected in triple negative breast cancer patients, and the latter were correlated with decreased survival. Our data suggest that PARP-1, in conjunction with *BRCA1*, could potentially be used as (a) biomarker(s) for patients’ stratification.

**Abstract:**

*BRCA1* and PARP are involved in DNA damage repair pathways. *BRCA1* mutations have been linked to higher likelihood of triple negative breast cancer (TNBC). The aim of the study was to determine PARP-1 expression and *BRCA1* mutations in circulating tumor cells (CTCs) of BC patients. Fifty patients were enrolled: 23 luminal and 27 TNBC. PARP expression in CTCs was identified by immunofluorescence. Genotyping was performed by PCR-Sanger sequencing in the same samples. PARP-1 expression was higher in luminal (61%) and early BC (54%), compared to TNBC (41%) and metastatic (33%) patients. In addition, PARP-1 distribution was mostly cytoplasmic in luminal patients (*p* = 0.024), whereas it was mostly nuclear in TNBC patients. In cytokeratin (CK)-positive patients, those with the CK^+^PARP^+^ phenotype had longer overall survival (OS, log-rank *p =* 0.046). Overall, nine mutations were detected; M1 and M2 were completely new and M4, M7 and M8 were characterized as pathogenic. M7 and M8 were predominantly found in metastatic TNBC patients (*p* = 0.014 and *p* = 0.002). Thus, PARP-1 expression and increased mutagenic burden in TNBC patients’ CTCs, could be used as an indicator to stratify patients regarding therapeutic approaches.

## 1. Introduction

Breast cancer (BC) is a multifactorial disease and accounts for 30% of cancers in women [1]. A variety of factors have been associated with its incidence, several of which are genetic. Indeed, approximately 40% of inherited cancers are due to mutations in the breast cancer susceptibility genes 1 and 2 (*BRCA1* and *BRCA2*) [2]. Recent estimates have shown that approximately 70% of patients with mutations in *BRCA1* and 45% in *BRCA2* will develop BC by the age of 70 [1,3].

Triple negative BC (TNBC) is the most aggressive BC subtype, devoid of hormone receptors (estrogen and progesterone) or human epidermal growth factor receptor 2 (HER2). TNBC is distinct and heterogeneous compared to the other subtypes [4]. Lack of targeted therapies for TNBC patients has led to an unmet need for new biomarkers [5,6]. TNBC is closely associated with *BRCA* mutations and especially germline mutations [7].

*BRCA1* is located on chromosome 17q21 and encodes a tumor suppressor protein of 1863 amino acid residues. *BRCA1* has been reported to have a plethora of roles in tumorigenesis from the start of a tumor, up to the regulation of epithelial-to-mesenchymal transition, cell motility, adhesion, invasion and ultimately metastasis. DNA damage and dysfunctions of repair are also features of cancer etiology [8]. *BRCA1* is involved in the DNA damage repair (DDR) processes, in particular the repair of double strand breaks (DSBs), hence safeguarding genomic stability and integrity [9,10].

Poly(ADP-ribose) polymerase 1 (PARP-1) enzyme is the most abundant and best characterized of the 17-member family in humans [11,12]. PARP-1 is a DNA damage sensor, also involved in DDR, primarily the repair of single-strand breaks (SSBs), but also DSBs [13,14]. Numerous studies have shown up-regulation of PARP-1 expression in cancer cell lines and patients’ tissues [15]. PARP-1 is mainly localized in the nucleus, but its cytosolic distribution has also been investigated [16,17,18].

The DDR process, where both *BRCA1* and PARP-1 are crucial, consists of five pathways. Repair of SSBs by PARP-1 is via the base excision repair pathway [11,12]. In the case of the more deleterious DSBs, repair is mediated by two complementary systems, namely the non-homologous end joining (NHEJ), which is more error-prone and can also depend on PARP activity, and the homologous recombination (HR) repair system, mediated amongst other DNA damage players by *BRCA1* [11,12,13]. Interestingly, more and more evidence suggests that PARP-1 is also essential in the HR process [19,20], supporting PARP’s complex roles and the interplay of the repair pathways. Administration of PARP inhibitors will result in the production of a SSB, which will be repaired by HR. In the HR-deficient (BRCA-mutated) cells, however, it will result in a DSB, when this SSB reaches the replication fork. Accumulation of DSBs will prove fatal for the cell as they will lead to cell apoptosis or accumulation of mutations with higher likelihood for cancer development [5,21,22,23,24].

New treatments in BC target the DDR [25]. In a very recent meta-analysis, PARP inhibitors have been shown to prolong progression free survival (PFS) and overall survival (OS) in patients with BRCA-mutated advanced BC [26]. Two PARP inhibitors, namely olaparib and talazoparib, have been approved as monotherapies for the management of locally advanced/metastatic HER2-negative BC, in patients with *BRCA1* or 2 germline mutations [6,22,23,27]. These inhibitors were identified following the OlympiAD [28,29] and EMBRACA [30,31] phase III trials, respectively. In the former study, approximately half of the patients had TNBC. The main outcome of both studies was the statistically significant improvement of PFS of BC patients, who were treated with the inhibitors compared to the control groups (7.0 vs. 4.2 months, HR = 0.58 for olaparib and 8.6 vs. 5.6 months, HR = 0.54 for talazoparib), without however a significant benefit on the OS (19.3 vs. 17.1 months for olaparib and 19.3 vs. 19.5 months for talazoparib).

Circulating tumor cells (CTCs) play an important role in the metastatic activity. The prognostic and predictive value of CTCs in BC is being constantly sought. In fact, CTCs’ detection and enumeration has been reported in early as well as metastatic BC and has been associated with poor clinical outcome (decreased PFS and OS) [32,33]. The epithelial marker, cytokeratin (CK) and particularly A45-B/B3 antibody (CK8, CK18, CK19) has been widely used for the characterization of cells as CTCs. Attention has lately been focused on the heterogeneity of CTCs and the identification of proteins/biomarkers and/or mutations in driver genes with prognostic and/or predictive significance [34,35,36].

The aim of the present study was to investigate PARP-1 expression and determine the existence of *BRCA1* mutations in BC patients’ CTCs, with the ultimate goal of identifying biomarkers capable of guiding informed decisions regarding patients’ treatment, especially those of the more aggressive subtypes.

## 2. Materials and Methods

### 2.1. Patients’ Samples and Cytospins’ Preparation

Fifty blood samples were obtained from BC patients, 23 of which were luminal and 27 were TNBC. Informed consent, which has been approved by the Ethics and Scientific Committees of our institution, was obtained from all subjects involved in the study.

Peripheral blood (10 mL in EDTA) was drawn from the middle of the vein puncture after discarding the first 5 mL in order to avoid contamination from skin epithelial cells during sample collection. Peripheral blood mononuclear cells (PBMCs) were isolated with Ficoll–Hypaque, after centrifugation at 1800 rpm for 30 min at 4 °C. PBMCs were washed twice with PBS and centrifuged at 1500 rpm for 10 min. Aliquots of 500.000 cells were centrifuged at 2000 rpm for 2 min on glass slides [37,38]. Cytospins were dried up and stored at −80 °C. In this study we didn’t use any magnetic isolation with EpCAM beads because the use of two different epithelial markers (EpCAM and CK) would decrease the recovery rate of CTCs. In addition, it has been shown that EpCAM is downregulated in many CTCs [39]. Instead, we followed the published methodology used in the past in many publications of our team [37,40,41,42].

### 2.2. Double Immunofluorescence

Double immunofluorescence staining experiments for CK/PARP were performed in patients’ cytospins (2 per patient), accompanied by control experiments (Appendix A) in cytospins with MCF-7 cells [37,38]. A PARP negative control was made by omitting the corresponding primary antibody, while including its secondary IgG antibody. Cells were initially fixed with 3% paraformaldehyde for 10 min and permeabilized with 0.5% Triton X-100 for 10 min, followed by blocking with 5% FBS for 1 h. PARP (1:200; Novus Biologicals, Littleton, CO, USA) was detected using anti-rabbit antibody labelled with Alexa Fluor 555 (ThermoFisher Scientific, Waltham, MA, USA). CK detection was achieved with the primary A45-B/B3 antibody (1:100; Amgen, Thousand Oaks, CA, USA) and its secondary anti-mouse antibody labelled with Alexa Fluor 488 (ThermoFisher Scientific, Waltham, MA, USA). Finally, cells were mounted on slides with 4′,6-diamidino-2-phenylindole (DAPI)-containing antifade reagent.

Cytospins were analyzed with the VyCAP system (VyCAP B.V., Enschede, The Netherlands) and a Leica TCS SP8 confocal microscope (Leica Microsystems, Wetzlar, Germany).

### 2.3. DNA Isolation from Cytospins

Cytospins with MCF-7 cells were used as positive controls to evaluate DNA isolation efficacy. Cells were removed from glass slides by scraping with a scalpel [43], after the staining procedure. The isolated cell pellet was centrifuged at 530× *g* for 10 min. Cells were resuspended in 200 μL PBS and then used for DNA extraction. gDNA was isolated with the High Pure PCR Template Preparation Kit (Roche, Mannheim, Germany) according to the manufacturer’s instructions for isolation of nucleic acids from mammalian whole blood or cultured cells. DNA purity and quantity were determined by absorbance readings at 260/280 nm with Quawell Q5000 UV-Vis Spectrophotometer (Quawell Technology, Inc., San Jose, CA, USA) and integrity was controlled by electrophoresis on 1% agarose gel.

### 2.4. Amplification and Preparation of PCR Products

PCR was performed using the C1000 Touch Thermal Cycler (BIO-RAD, Hercules, CA, USA). The following primers embracing *BRCA1* exon 20 were used [44]:

Forward Primer: 5′ ATATGACGTGTCTGCTCCAC 3′.

Reverse primer: 5′ CTGCAAAGGGGAGTGGAATAC 3′.

The amplification mixture of a total volume of 25 μL included the isolated gDNA as the template and final concentrations of 1x PFU DNA polymerase, 1× supplied buffer, 1.5 mM MgCl_2_ and 200 μΜ dNTPs. Final primer concentrations were 0.4 μΜ. The cycling protocol consisted of pre-incubation at 94 °C for 5 min, followed by 35 cycles of denaturation at 94 °C for 30 s, annealing at 56 °C for 30 s, extension at 72 °C for 30 s, and a final extension at 72 °C for 3 min. Confirmation of the desired PCR products (232 bp) was obtained by electrophoresis on 1.5% agarose gel.

The NucleoSpin Gel and PCR clean-up kit (Macherey–Nagel, Düren, Germany) was used to purify the PCR products following the manufacturer’s instructions. Purification was confirmed by electrophoresis on 1.5% agarose gel.

### 2.5. Genotyping—Sanger Sequencing

The purified PCR products were sequenced at CEMIA SA (Larissa, Greece). The sequence of the forward primer used for Sanger sequencing was 5′ ATATATGACGTGTCTGCTCCAC 3′. Display and analysis of the chromatograms (Appendix A) was done using the FinchTV program. Alignment of the sequences from the chromatograms was done through the ClustalW2 program using the corresponding sequence from NCBI (NG_005905.2:160741-161041 Homo sapiens *BRCA1* DNA repair associated (BRCA1), RefSeqGene (LRG_292) on chromosome 17).

Exon 20 of *BRCA1* was chosen as focus was on the detection of the mutations c.5366dupC, R1751X and G1738R which are the most frequent in the Greek population with BC. The intronic mutations were found, because the primers used in PCR included both at the 5′ and 3′ some part of intronic regions flanking exon 20.

### 2.6. Statistical Analysis

Spearman analysis, Mann–Whitney test, Kruskal–Wallis and χ^2^ was used to compare the detected mutation with the expression of PARP-1, as well as available clinical data. Kaplan–Meier survival tests for OS were performed. All analyses were performed on IBM SPSS statistics version 26 software (IBM, Armonk, NY, USA). A value of *p* ≤ 0.05 was used to identify significant results.

## 3. Results

### 3.1. PARP-1 Expression in the CTCs of BC Patients

From the 50 samples which were analysed from BC patients, 23 were luminal and 27 were TNBC. Among BC subtypes, the phenotype of CK^+^PARP^+^ was present in 14 out of 23 (61%) luminal and 11 out of 27 (41%) TNBC patients (Figure 1A). Regarding disease status, CK^+^PARP^+^ CTCs were detected in 19 out of 35 (54%) early BC patients compared to 5 out of 15 (33%) metastatic patients (Figure 1B and Appendix A).

Regarding BC subtype, incidence of the CK^+^PARP^+^ phenotype among patients’ CTCs was 91% in luminal versus 84% in TNBC patients (Figure 1C). In terms of disease status, incidence of the CK^+^PARP^+^ phenotype was 91% in early BC versus 90% in metastatic (Figure 1D and Appendix A). Hence PARP-1 expression was found higher in luminal and early BC patients.

Concentrating on the status of the disease, the CK^+^PARP^+^ phenotype was present in 44% of early (8 out of 18) and 33% of metastatic (3 out of 9) TNBC patients, while it was present in 58% (11 out of 19) of early and 50% (2 out of 4) of metastatic luminal patients (Appendix A). The CK^+^PARP^−^ phenotype was present in 11% of both early and metastatic TNBC patients (2 out of 18 and 1 out of 9, respectively), as well as in 16% of early luminal patients (3 out of 19), but not in the metastatic luminal patients (Appendix A).

Incidence of the CK^+^PARP^+^ phenotype was 83% among both early and metastatic TNBC patients’ CTCs. In terms of the luminal patients, 89% of the CTCs of early patients and all CTCs of metastatic patients corresponded to the CK^+^PARP^+^ phenotype (Appendix A). Based on the status of the disease, PARP-1 expression was again higher (although not statistically significant) in luminal and early BC patients.

Two different patterns of PARP expression were observed based on its subcellular localization: nuclear and cytoplasmic (Figure 2).

Nuclear expression of PARP-1 was observed in 30% of both the TNBC and luminal patients (8 out of 27 and 7 out of 23, respectively), while cytoplasmic expression was seen in 15% (4 out of 27) of TNBC and 43% (10 out of 23) of the luminal patients; the latter observation was statistically significant (*p* = 0.024, Figure 3A and Appendix A).

Incidence of the nuclear subcellular localization among patients’ CTCs was 61% in TNBC compared to 32% in luminal and accordingly that of the cytoplasmic was 39% and 68%, respectively (Figure 3B and Appendix A). PARP-1 was therefore predominantly found in the cytoplasm of luminal patients’ CTCs, whereas it was mostly found in the nucleus of TNBC patients.

Data were analyzed based on the disease status (Appendix A). Nuclear expression of PARP-1 was observed in 33% (6 out of 18) of the early and 22% (2 out of 9) of the metastatic TNBC patients, while it was in 10% and 25% (2 out of 19 and 1 out of 4, respectively) of the early and metastatic luminal patients, respectively (Appendix A). Cytoplasmic expression of PARP-1 was observed in 17% and 11% of early and metastatic TNBC patients (3 out of 18 and 1 out of 9, respectively), while it was 47% and 25% (*p =* 0.035) in early and metastatic luminal patients (9 out of 19 and 1 out of 4, respectively; Appendix A).

Incidence of the nuclear subcellular localization among patients’ CTCs was 73% and 67% in early and metastatic TNBC patients, respectively, whereas it was 43% and 50% in early and metastatic luminal patients, respectively (Appendix A). The incidence of the cytoplasmic subcellular localization among patients’ CTCs was 27% and 33% in early and metastatic TNBC patients, respectively, whereas it was 57% and 50% in early and metastatic luminal patients, respectively (Appendix A). Results further confirmed that PARP-1 in luminal patients was mostly cytoplasmic, while in TNBC patients was mostly nuclear.

### 3.2. BRCA1 Mutations Identified in CTCs of BC Patients

Nine mutations of *BRCA1* were detected from the sequencing process in the same patients’ samples (Table 1). Two, namely c.5277 + 65C > T and c.5277 + 67T > C were new, not previously identified mutations, whereas seven (M3-M9) had been previously identified and characterized (NCBI; https://www.ncbi.nlm.nih.gov/snp/, accessed on 13 September 2021) [45,46]. All of the detected mutations were found in the heterozygous state.

Of the nine identified mutations, one (M1) was higher in luminal patients, four (M2, M4, M6, M9) were found in both luminal and TNBC patients at comparable levels and four (M3, M5, M7, M8) were mostly found in TNBC patients (Figure 4A). Two of the three pathogenic mutations, namely M7 and M8, were mostly detected in TNBC patients. More specifically, M7 was exclusively found in TNBC patients (19%; 5 out of 27), a result which was statistically significant (*p* = 0.03, Figure 4A). M8 was found in 15% of TNBC (4 out of 27) compared to 9% of luminal (2 out of 23) patients (Figure 4A and Appendix A). Interestingly, the majority of mutations were present in TNBC patients even when CK-positive TNBC patients were compared to CK-positive luminal patients.

Between early and metastatic BC patients, results showed higher prevalence of the mutations in TNBC patients as reported above, with the percentages for the detected mutations being presented in Figure 4B and Appendix A. More precisely, M7 and M8 were mostly detected in metastatic TNBC patients in a statistically significant manner (*p* = 0.014 and *p* = 0.002, respectively).

When analysis was performed only in CK-positive patients, as was the case earlier, the majority of mutations were present in CK-positive TNBC patients, as compared to CK-positive luminal patients, and more specifically in metastatic TNBC patients. Indeed, only TNBC patients had the M3, M5, M6, M7 and M8 mutations in their CTCs or mainly the TNBC patients had the M4 and M1 mutations.

### 3.3. BRCA1 Mutations in CTCs of BC Patients and Clinical Outcome

Clinical data regarding follow up (OS and PFS) were available for 24 patients (7 out of 23 luminal and 17 out of 27 TNBC). After a median follow-up period of 25 months for luminal and 50 months for TNBC patients, 1 and 9 deaths, respectively, were registered as a consequence of disease progression.

Among the CK-positive early BC patients, the CK^+^PARP^+^ phenotype was associated with a longer OS (log-rank *p* = 0.046, HR = 1.37; Appendix A).

Interestingly, these results are supported by data provided by KMplot, an online tool [47], whereby basal-like BC patients with higher expression of PARP-1 have been indicated to have a longer disease-free interval (DFI; log-rank *p* = 0.022, HR = 0.67; Appendix A) and OS (log-rank *p* = 0.015, HR = 0.59; Appendix A).

Focusing on the pathogenic mutations, M4, M7 and M8 exhibited correlation with TNBC patients’ OS (log-rank *p* = 0.008, HR = 15.07 for M4, *p* = 0.019, HR = 6.4 for M7 and *p* = 0.019, HR = 6.4 for M8; Figure 5).

## 4. Discussion

In the present study we investigated the expression and subcellular localization of PARP-1 in CTCs of luminal and TNBC patients. In addition, we detected *BRCA1* mutations in these CTCs and assessed their relationship with severity of disease and clinical outcome.

Based on BC subtype, expression of PARP-1 was identified in 61% of luminal and 41% of TNBC patients. Based on the disease status, 54% and 33% of patients with early and metastatic BC, respectively, expressed PARP-1. Notably, we have found that PARP-1 expression is higher in patients with less severe disease status, i.e., early stage and luminal patients. The percentage of PARP-1 expression in tissue microarrays so far is approximately 24–33% [17,48]. In addition, after immunohistochemistry of biopsies/tissue microarrays, high (cytoplasmic) PARP expression was detected in 36% of TNBC tumors [17]. Furthermore, PARP-1 mRNA as well as protein expression has been found to be decreased in luminal compared to HER2-enriched and basal BC tumors [49]. The role of PARP-1 expression in tumor grade is ambiguous; high tumor grade has been reported to correlate with low (*p* = 0.003) [16] or high [17,48] PARP-1 expression in BC. Furthermore, high PARP-1 expression has been associated with a more advanced clinical stage [16]. To the best of our knowledge this is the first study that evaluates PARP-1 expression in patients’ CTCs.

Examination of PARP-1 localization in BC patients showed that PARP-1 was distributed either in the nucleus or in the cytoplasm of CTCs. It has been shown that PARP-1 has a nuclear localization signal in its N-terminal DNA-binding domain [50]. In fact, the presence of PARP-1 in the nucleus is essential for the maintenance of genomic integrity and cell survival. Hence the importance of the nuclear localization for PARP functions is well established, whereas the cytoplasmic localization has equally gained interest [18]. Therefore, it was noteworthy that in our study the nuclear localization was more prominent in TNBC patients’ CTCs (61%), whereas cytoplasmic localization was mostly seen in luminal patients (43%, *p =* 0.024), showcasing a higher percentage of CTCs with cytosolic PARP-1. These results imply that PARP-1 was potentially more active in TNBC patients. In line to this observation, our data based on disease status, confirmed nuclear expression of PARP-1 in early and metastatic TNBC patients’ CTCs. Similar to our observations, distribution of PARP-1 in the two subcellular compartments has been previously shown in tissues of different subsets of BC patients [16,17]. PARP-1 expression has been previously determined in the majority (82%) of BRCA1-dependent BC cases, with a predominantly nuclear subcellular localization [51].

Although this is a pilot study, and the cohort of patients is rather small it was interesting that the CK-positive early BC patients with the CK^+^PARP^+^ phenotype had longer OS compared to the negative patients (log-rank *p* = 0.046). KMplot database (https://kmplot.com/analysis/, accessed on 27 September 2021), further supported our data, by demonstrating that high PARP-1 expression was correlated with longer DFI (log-rank *p* = 0.022, HR = 0.67) and OS (log-rank *p* = 0.015, HR = 0.59) in basal-like BC patients [47]; TNBC is classified as a subtype of basal-like BC. However, a meta-analysis has shown that high PARP expression was associated with poor OS in BC patients (HR = 1.38, 95% CI = 1.28–1.49, *p* < 0.001) [52].

One of the main objectives of this study was also to detect *BRCA1* mutations in CTCs and not in plasma, to directly compare mutations in cancer cells and PARP expression. Furthermore, the origin of these mutations in plasma is not defined (normal or tumor cells). This approach is giving a better profile of CTCs regarding the DNA repair system. In addition, it is well known that some mutations do not exist in the primary tumor and can be detected in CTCs at a distinct time point during cancer evolution [53]. Furthermore, it is possible that if the primary tumor is removed and the number of CTCs is very low, the “de novo” mutations could not be detected in the plasma. In addition, our approach is an easy assay with one isolation step of CTCs (Ficoll density gradient).

Nine *BRCA1* mutations were detected in BC patients’ CTCs. The majority of mutations was observed in the CTCs of TNBC patients compared to luminal patients, suggesting that mutational burden tends to be higher in TNBC patients. This coincides with the results of a meta-analysis suggesting that patients with *BRCA1* mutations will more probably have more TNBC and higher tumor burden [54].

Of the identified pathogenic mutations, M7 was exclusively detected in TNBC patients (19%, *p* = 0.03), while M8 was mostly found in TNBC patients (double the percentage of that seen in luminal patients). M7 is a splice acceptor mutation (NCBI; https://www.ncbi.nlm.nih.gov/snp/rs80358173, accessed on 13 September 2021) and is identified as a pathogenic mutation with Combined Annotation-Dependent Depletion (CADD) > A:34, G:34, C:34, belonging to 0.1% of the most harmful SNPs. M8 is a missense variant with CADD > G:26.8, which means that it belongs to the 1% of the most harmful SNPs. Among the TNBC patients, M7 and M8 were detected in more metastatic patients (44% for both, *p* = 0.014 and *p* = 0.002, respectively) compared to early patients. M4 (A > T mutation) (https://www.ncbi.nlm.nih.gov/snp/rs80358069, accessed on 13 September 2021) is a splice acceptor variant. In Chevalier et al., the reported A > G mutation (c.5194-2A > G) caused a change in the splice acceptor region which affects alternative splicing and is thus identified as pathogenic [46]. TNBC patients bearing the M4 (*p* = 0.008), M7 (*p* = 0.019) and M8 (*p* = 0.019) mutations were also correlated with decreased OS. These observations tend to agree with the characterization of the mutations as pathogenic and suggest that they can indeed be of particular interest. However, our cohort is small, and these results are only indicative of the severity of these mutations.

Recently olaparib has been approved by the FDA for adjuvant treatment of BRCA-mutated HER2-negative high-risk early-stage BC patients (phase III OlympiAD trial). Our approach can give a real time observation of *BRCA1* and PARP expression in cancer cells, providing an interesting tool for stratifying patients that could benefit from this target therapy. Furthermore, our analysis revealed that among the BC patients the majority of *BRCA1* mutations were observed in TNBC patients, who also expressed nuclear PARP-1. This is rather interesting as it would imply that patients with high mutation burden and PARP-1 nuclear expression would be more likely to benefit from current regimens with PARP inhibitors.

## 5. Conclusions

TNBC is associated with limited treatment options. Use of liquid biopsy and of CTCs can offer the advantage of profiling patients in real time and identifying subpopulations which can benefit from a specific therapy.

To the best of our knowledge, this is the first study addressing PARP-1 expression in CTCs. Results of this study indicate that (high) nuclear PARP-1 expression and detection of *BRCA1* mutations are characteristics of TNBC patients’ CTCs, providing useful biomarkers that are also potentially relevant to patients’ outcome.

## Figures and Tables

**Figure 1 cancers-14-01731-f001:**
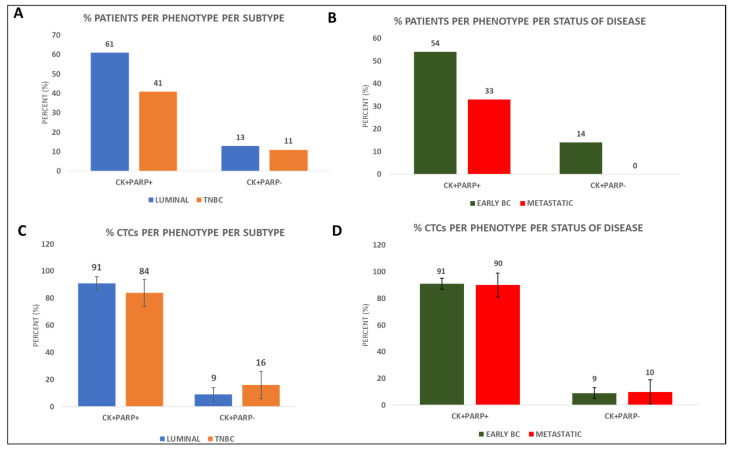
PARP-1 expression in BC patients’ CTCs. (**A**) Percentage of luminal and TNBC patients with CK^+^PARP^+^ or CK^+^PARP^−^ phenotypes; (**B**) Percentage of early and metastatic BC patients with the corresponding CTC phenotypes; (**C**) Percentage of CTCs with CK^+^PARP^+^ or CK^+^PARP^−^ phenotypes in luminal and TNBC patients; (**D**) Percentage of CTCs with CK^+^PARP^+^ or CK^+^PARP^−^ phenotypes in early and metastatic BC patients.

**Figure 2 cancers-14-01731-f002:**
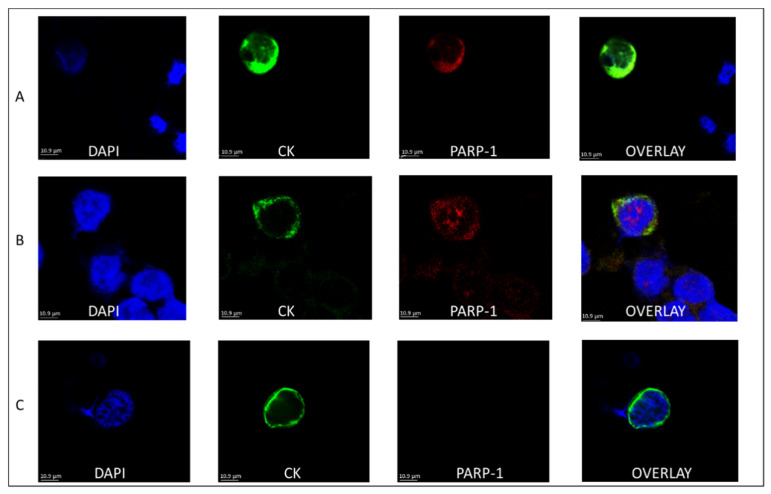
Subcellular localization of PARP-1 in BC patients’ CTCs. The first column represents nuclei stained with DAPI, the second column represents cells expressing CK, the third cells expressing PARP-1 and the fourth represents the overlay (DAPI/CK/PARP-1). Representative panels of cells with (**A**) Cytoplasmic expression of PARP-1; (**B**) Nuclear expression of PARP-1 and (**C**) No expression of PARP-1 are shown. Scale bars = 10 μm.

**Figure 3 cancers-14-01731-f003:**
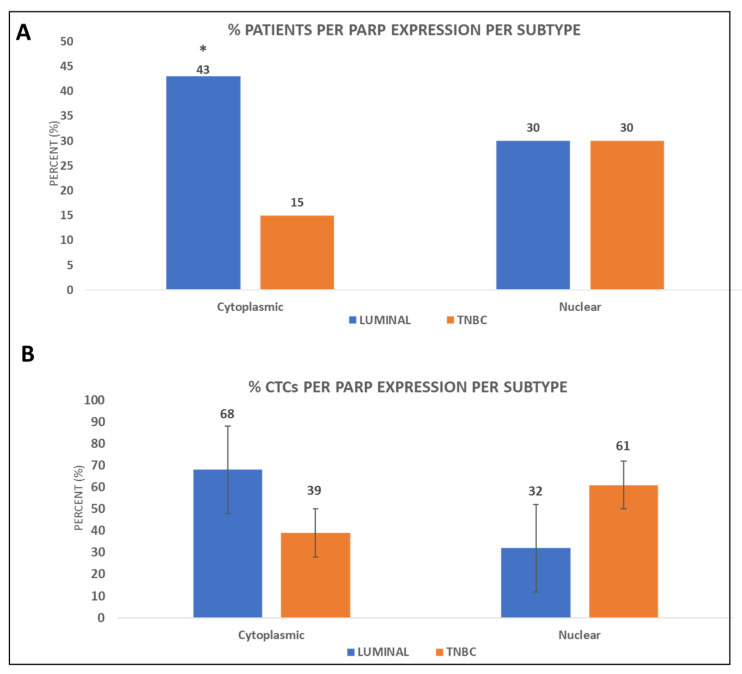
PARP-1 subcellular distribution in BC patients’ CTCs. (**A**) Percentage of luminal and TNBC patients with either cytoplasmic or nuclear expression of PARP-1 in their CTCs (* *p* = 0.024); (**B**) Percentage of CTCs with cytoplasmic or nuclear PARP-1 localization in luminal and TNBC patients.

**Figure 4 cancers-14-01731-f004:**
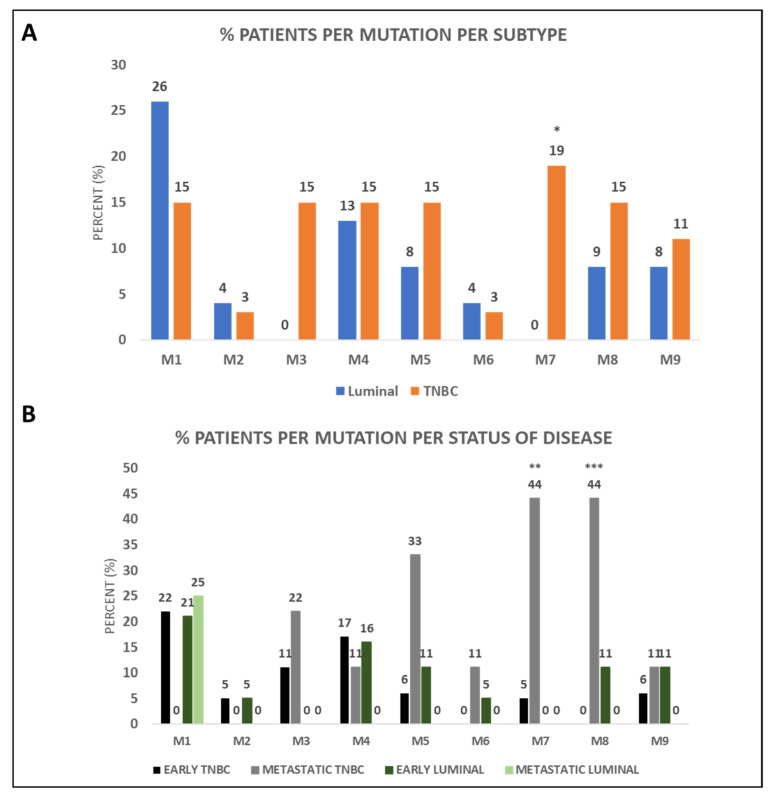
*BRCA1* mutations in BC patients’ CTCs. (**A**) Percentage of luminal and TNBC patients bearing the mutations M1 to M9 (* *p* = 0.03); (**B**) Percentage of early and metastatic TNBC and luminal patients bearing the mutations M1 to M9 (** *p* = 0.014, *** *p* = 0.002).

**Figure 5 cancers-14-01731-f005:**
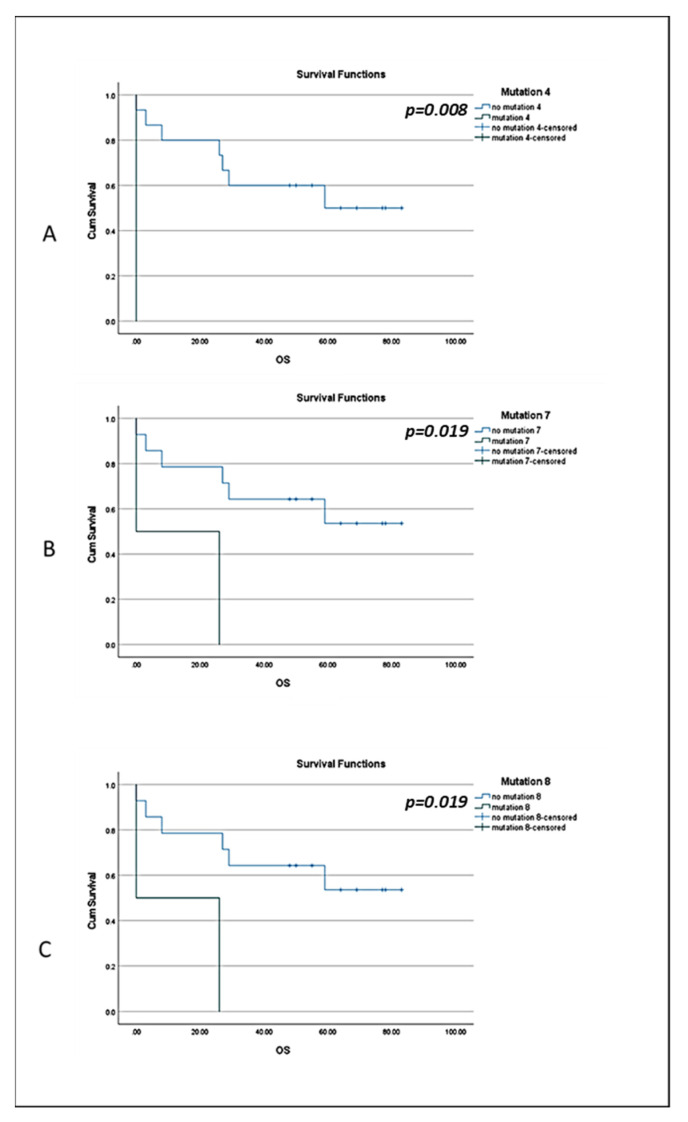
Estimates of OS of TNBC patients in respect to the presence of the identified mutations. Kaplan–Meier survival curve for TNBC patients with the mutations (**A**) M4 (*p* = 0.008); (**B**) M7 (*p* = 0.019); (**C**) M8 (*p* = 0.019) in their CTCs.

**Table 1 cancers-14-01731-t001:** Mutations detected in *BRCA1* and their biological significance.

Number	Mutation	rs Code	Number of Patients	Exon/Intron	Biological Effect
M1	G > A	c.5277 + 65C > T	10	Intron	Unknown ^1^
M2	A > G	c.5277 + 67T > C	2	Intron	Unknown ^1^
M3	A > T	rs2051507989	5	Exon	Synonymous
M4	A > T	rs80358069	7	Intron	Pathogenic
M5	G > C	rs1567764460	6	Exon	Missense Variant
M6	T > C	rs1057524456	2	Intron	No Information
M7	G > T	rs80358173	5	Intron	Pathogenic
M8	G > C	rs377595653	6	Exon	Pathogenic
M9	A > C	rs80357270	5	Exon	No functional impact

^1^ OPEN CRAVAT website didn’t report any biological effect regarding these variants.

## Data Availability

Data presented in this study are available upon request from the corresponding authors.

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
