# Peer review of "PARP-1 Expression and BRCA1 Mutations in Breast Cancer Patients’ CTCs"

_cancers, 2022, doi:10.3390/cancers14071731_

Round 1

Reviewer 1 Report

In this manuscript Sklias et al. investigated PARP-1 expression and BRCA1 mutations in circulating tumor cells of breast cancer patients. They analyzed CTCs by detecting cytokeratin-positive cells among PBMCs. They observed a high PARP1 expression that localizes to nuclei in case of TNBC, unlike cytoplasmic localization in case of luminal cancers. The authors also analyzed mutations in BRCA1 gene after extracting DNA from cytospin slides. Finally, they analyzed whether the specific mutations correlated with disease subtype or disease outcome.     

Overall, this is a well-planned pilot study yielding meaningful results. Although a lot more data from the patients will be needed to advance the proposed tests in the clinic, I believe that the manuscript is suitable for publication in Cancers. I have a few minor points that authors may like to address. These points pertain to feasibility of translation in clinic, and comparison with competing technologies.

Minor points:

  1. The authors use cytokeratin positivity as in indicator of breast epithelial cells. They must be aware that many others use additional marker e.g., EpCAM for identifying CTCs (which is also utilized in FDA-approved CellSearch method in United States and elsewhere). A brief explanation as to why this will not be a problem would help a reader.
  2. A popular method of mutation detection is identifying these mutations in plasma. Although it makes sense to detect mutations directly in the cytospin that has CTCs, as the authors did in this manuscript, does it offer a distinct advantage as compared to mutation detection in plasma? I am thinking from the feasibility perspective in the clinic. A brief explanation may be good.
  3. The authors may like to discuss a bit more, with specific examples, whether performing both BRCA1 mutation analysis and PARP1 staining would improve decision making in the clinic? Will there be specific scenarios whether a PARP inhibitor would be offered or not offered based on these analyses singly or in combination?  

Author Response

Reviewers’ comments

Reviewer 1:

  1. The authors use cytokeratin positivity as in indicator of breast epithelial cells. They must be aware that many others use additional marker e.g., EpCAM for identifying CTCs (which is also utilized in FDA-approved CellSearch method in United States and elsewhere). A brief explanation as to why this will not be a problem would help a reader.

We thank the reviewer for the fruitful comments. It is true that the CellSearch system isolates tumor cells using EpCAM-coated magnetic beads and detects CTCs based on CK. However, it is now widely accepted that this approach has very low recovery rate, especially in adjuvant setting, due to the EMT nature of CTCs. Therefore, we avoided to use magnetic isolation with EpCAM, because the use of two different epithelial markers would decrease the recovery rate of CTCs. In addition, it has been shown that EpCAM is downregulated in many CTCs (Sieuwerts et al., 2009). Instead, we followed the published methodology used in the past in many publications of our team (Kallergi et al., 2008, 2011; Kallergi, Konstantinidis, et al., 2013; Kallergi, Mavroudis, et al., 2007), which is based on Ficoll Density Gradient and Cytokeratin (8, 18, 19) detection. We have now included this clarification in the revised version page 5.

  1. A popular method of mutation detection is identifying these mutations in plasma. Although it makes sense to detect mutations directly in the cytospin that has CTCs, as the authors did in this manuscript, does it offer a distinct advantage as compared to mutation detection in plasma? I am thinking from the feasibility perspective in the clinic. A brief explanation may be good.

The reviewer is right in suggesting that the identification of mutations in the plasma for different genes such as BRCA1, EGFR etc is a very easy procedure and popular assay in many types of cancer. However, in the current study we wanted to detect mutations in patients’ tumor cells. Therefore, we used samples enriched with CTCs. In plasma the origin of these mutations is not defined (normal or tumor cells). This approach is giving a better profile of CTCs regarding the DNA repair system. In addition, it is well known that some mutations do not exist in the primary tumor and can be detected in CTCs at a distinct time point during cancer evolution (Yu et al., 2014). Furthermore, it is possible that if the primary tumor is removed and the number of CTCs is very low the “de novo” mutation could not be detected in the plasma. In addition, our approach is an easy assay with one isolation step of CTCs (Ficoll density gradient). Using this approach, we can detect any mutation (both germline and de novo) in cancer cells at any time point and we can directly compare this to PARP expression in the same patients’ samples. These views are now reflected in the discussion section in page 15

  1. The authors may like to discuss a bit more, with specific examples, whether performing both BRCA1 mutation analysis and PARP1 staining would improve decision making in the clinic? Will there be specific scenarios whether a PARP inhibitor would be offered or not offered based on these analyses singly or in combination?

We thank the reviewer for the comment. Based on our approach we provide a useful real time tool for stratifying patients that could benefit from PARP inhibitors. For example, patients with pathogenic mutations in BRCA1 and/or PARP nuclear expression could potentially be treated with PARP inhibitors. This perception is now discussed in page15-16.

 Sincerely,

Galatea Kallergi

Reviewer 2 Report

In this report Sklias et al correlate the geography of PARP-1 cellular expression and certain BRCA1 mutations with aggressiveness of breast cancers. The work has some merit. However, this reviewer believes that at this stage the work is too incipient for publication. The descriptions of certain results and conclusions are confusing. Further, some data are presented in a way that is hard to read or analyze. An overall re-organization/re-write/re-analysis is warranted before this work is ready for publication. Please see below my specific comments.

Major comments

What do the error bars in Figure 1C, and D represent and why are there no bars in A and B? You state that a statistical test (chi square) was done to determine whether the differences in the different values in the panels in Fig1 is statistically significant.  Then what are the error bars? What test was done for those? Please include a table in the supplemental material with the raw numbers and the test and p-values for all calculations. As it stands, it is confusing and is hard to interpret these data.

Figure 2 does not appear to show anything new that is not shown in Fig.1. The data seem to just be re-organized differently. I suggest either combining Figs 1 and 2 or relegating one to the supplemental file. Statistical tests mentioned above should be performed for both supplemental and main text figures.

For figure 3, please be specific on what the overlay represents (e.g. DAPI, CK and PARP-a). Also, I am not convinced by the panels showing cytoplasmic expression of CK and PARP-1 (Fig. 3A). First of all, the DAPI signal is almost non-visible in panel A. Second, why is there no DAPI signal in the cell with CK and PARP-1 expression? Does this cell not have a nucleus? In order to be convinced that there is cytoplasmic expression, I would have to see the nucleus of this cell (by DAPI staining). Otherwise, one could argue that the reason you think is cytoplasmic is because you failed to stain the nucleus. Please indicate in the figure, where is the nucleus of this cell, perhaps by drawing a contour of the cell. As far as I am concerned, no conclusion can be drawn about cytoplasmic staining from the panels presented.

On the same topic as above, Fig4 is inconclusive. The error bars overlap so one could argue that there is no difference between nuclear and cytoplasmic. A better way of representing the data when there is a lot of noise is with box plots.  

Line 227: “A similar observation was made when data were analyzed based on the disease status (Figure 5)”. It does not seem that the observation was similar. There are no p-values for Fig.5 and the error bars overlap. Again, it seems that these data are inconclusive.

The identification of two new mutations in BRCA1 is intriguing but the biological effect is missing. It is possible to determine the effect using available machine learning algorithms such as FATMHH, VEST and CHASM scores. Please run your mutations through some of these algorithms using the OPEN CRAVAT website. Also, I don’t understand the nomenclature of this mutation. The standard nomenclature for intronic mutations is by adding a + or – to the last base of the preceding or succeeding exon (e.g. c.3430-6C>T). Additionally, a “c.” should be put in front of the mutation. You do this at line 353 for the reported mutation by Chevalier et al. Please do the same here. Please see this paper (PubMed ID 17251329) for accepted standard mutation nomenclature or this website: https://www.hgvs.org/mutnomen/refseq.html#standard. Also, please note that the nomenclature is OK for the ones that have an rs number because the reader can check them on NCBI using that reference number.

How were the mutations described identified? In your materials and methods, you only sequence Exon 20. Why did you focus on Exon 20 and not sequence the whole gene?  If so, how do you find intronic mutations? It is unclear how this was done. Please explain in more details.

Minor comments

This statement is awkward (lines 78-81): “Synthetic lethality, induced by PARP inhibitors, in cells with mutations in genes involved in HR repair (such as BRCA1 and BRCA2) stems from their inability to repair SSBs. Such breaks will result in cessation of copying and will be eventually transformed into DSBs in need of HR for them to be repaired.” Although BRCA1 does have a role in SSB processing, BRCA2 is primarily DSB because it recruits RAD51. Also, how is an SSB converted into a DSB and what do you mean by “cessation of copying”? Do you mean that it will be converted by the replication fork? Please expand the explanation for this statement.

This statement is also generally incorrect (lines 83-85): “Accumulation of DSBs beyond an acceptable threshold will prove fatal for the cell as they will lead to cell apoptosis or accumulation of mutations with higher likelihood for cancer development [5, 85 21-24].” What do you mean by “beyond an acceptable threshold”? One unrepaired double strand break is lethal in most instances. What you probably mean is that too many DSBs overwhelm the DDR and will kill the cells because some will not be repaired. Please rephrase.

Why are BRCA1 and BRCA2 italicized but PARP1 is not?

Please define CK in both introduction and abstract. Also describe this gene in the introduction.

Please label Y axes for Figs 1, 2, 4, 5 and 6 with “Percent”.

Author Response

Reviewer 2:

  1. What do the error bars in Figure 1C, and D represent and why are there no bars in A and B? You state that a statistical test (chi square) was done to determine whether the differences in the different values in the panels in Fig1 is statistically significant. Then what are the error bars? What test was done for those? Please include a table in the supplemental material with the raw numbers and the test and p-values for all calculations. As it stands, it is confusing and is hard to interpret these data.

The authors would like to thank the reviewer for the meticulous review of the statistical analysis, as this will help clarify certain aspects, allow proper interpretation of the data and will enhance the credibility of the observations of the study.

Graphs 1A and 1B represent percentage (%) of patients, hence error bars are not depicted. On the contrary, graphs 1C and 1D represent the mean percentage (%) of CTCs; these are scale parameters and analysis is accompanied by error bars.

The analysis performed for graphs 1A and 1B is indeed Chi-square, while Mann-Whitney analysis was performed for the quantitative data of graphs 1C and 1D. This explanation is now added to page 6.

Tables S1 and S2 are now included in the Supplementary data to clarify the statistical tests performed and the statistical significance of the data.

  1. Figure 2 does not appear to show anything new that is not shown in Fig.1. The data seem to just be re-organized differently. I suggest either combining Figs 1 and 2 or relegating one to the supplemental file. Statistical tests mentioned above should be performed for both supplemental and main text figures.

Figure 2 was showing the distribution of different phenotypes in adjuvant vs metastatic setting in every different subtype. However, according to the reviewer’s suggestion Figure 2 is now part of the supplementary data, as Figure S3 (removed from page 9). Figures in the main text and in supplementary data have been re-numbered. Table S1 is also included in the Supplementary data to clarify the statistical tests performed and the statistical significance of the data.

  1. For figure 3, please be specific on what the overlay represents (e.g. DAPI, CK and PARP-a). Also, I am not convinced by the panels showing cytoplasmic expression of CK and PARP-1 (Fig. 3A). First of all, the DAPI signal is almost non-visible in panel A. Second, why is there no DAPI signal in the cell with CK and PARP-1 expression? Does this cell not have a nucleus? In order to be convinced that there is cytoplasmic expression, I would have to see the nucleus of this cell (by DAPI staining). Otherwise, one could argue that the reason you think is cytoplasmic is because you failed to stain the nucleus. Please indicate in the figure, where is the nucleus of this cell, perhaps by drawing a contour of the cell. As far as I am concerned, no conclusion can be drawn about cytoplasmic staining from the panels presented.

This figure is now Figure 2. The legend is changed to “….and the fourth represents the overlay (DAPI/CK/PARP-1).” to specify what the overlay stands for (page 8-9).

The quality of the images greatly depends on the resolution of the different screens and/or printing quality. Panel A has been changed, according to reviewer’s suggestion, with a more representative image depicting clearly the cytoplasmic localization of PARP-1 (page 8).

  1. On the same topic as above, Fig4 is inconclusive. The error bars overlap so one could argue that there is no difference between nuclear and cytoplasmic. A better way of representing the data when there is a lot of noise is with box plots.

Figure 4 is now Figure 3. We value the reviewer’s comment and understand the rationale behind it. We have attempted to present data in box plots. However, they do not ameliorate the graph. Indeed, error bars are high, but represent the acquired data. Table S1 is also included in the Supplementary data to clarify the statistical tests performed and the statistical significance of the data.

  1. Line 227: “A similar observation was made when data were analyzed based on the disease status (Figure 5)”. It does not seem that the observation was similar. There are no p-values for Fig.5 and the error bars overlap. Again, it seems that these data are inconclusive.

Figure 5 was showing the distribution of PARP in the cell (cytoplasmic or nuclear) in adjuvant vs metastatic setting in every subtype.  However, according to the reviewer’s suggestion Figure 5 is relegated to the supplementary data as Figure S4. “A similar observation was made” is deleted. Graph B has not been reformatted into box plots, as they do not offer more clarity. Table S1 is also included in the Supplementary data to clarify the statistical tests performed and the statistical significance of the data.

  1. The identification of two new mutations in BRCA1 is intriguing but the biological effect is missing. It is possible to determine the effect using available machine learning algorithms such as FATMHH, VEST and CHASM scores. Please run your mutations through some of these algorithms using the OPEN CRAVAT website. Also, I don’t understand the nomenclature of this mutation. The standard nomenclature for intronic mutations is by adding a + or – to the last base of the preceding or succeeding exon (e.g. c.3430-6C>T). Additionally, a “c.” should be put in front of the mutation. You do this at line 353 for the reported mutation by Chevalier et al. Please do the same here. Please see this paper (PubMed ID 17251329) for accepted standard mutation nomenclature or this website: https://www.hgvs.org/mutnomen/refseq.html#standard. Also, please note that the nomenclature is OK for the ones that have an rs number because the reader can check them on NCBI using that reference number.

We thank the reviewer for the comment. We are providing the primers so it is easier to check the sequence in NCBI blast to find each new mutation in the sequence. The naming of the novel mutations was carried out according to this sequence.  Further, according to the reviewer suggestion, we have used the OPEN CRAVAT website and it didn’t report any biological effect regarding the two new variants. It was only reported that they were intronic variants. According to OPEN CRAVAT the mutation G229A could be named c.5277+65C>T and A231G could be named c.5277+67T>C. We have now made the corresponding changes in the text (pages 10).

  1. How were the mutations described identified? In your materials and methods, you only sequence Exon 20. Why did you focus on Exon 20 and not sequence the whole gene? If so, how do you find intronic mutations? It is unclear how this was done. Please explain in more details.

The mutations described were identified through alignment in clustalW2. We focused on exon 20 of BRCA1 because we focused on the detection of the mutations c.5366dupC, R1751X and G1738R which are the most frequent in the Greek population with breast cancer. We have now clarified this in the revised version’s methods in page 6. The intronic mutations were found because the primers used in PCR included both at the 5’ and 3’ some part of intronic regions flanking exon 20.

  1. This statement is awkward (lines 78-81): “Synthetic lethality, induced by PARP inhibitors, in cells with mutations in genes involved in HR repair (such as BRCA1 and BRCA2) stems from their inability to repair SSBs. Such breaks will result in cessation of copying and will be eventually transformed into DSBs in need of HR for them to be repaired.” Although BRCA1 does have a role in SSB processing, BRCA2 is primarily DSB because it recruits RAD51. Also, how is an SSB converted into a DSB and what do you mean by “cessation of copying”? Do you mean that it will be converted by the replication fork? Please expand the explanation for this statement.

For clarity the phrase is changed to: “Administration of PARP inhibitors will result in the production of a SSB, which will be repaired by HR. In the HR-deficient (BRCA mutated) cells, however it will result in a DSB when this SSB reaches the replication fork”. According to the reviewer suggestion we have now rephrased the above text in the revised version (page 3).

  1. This statement is also generally incorrect (lines 83-85): “Accumulation of DSBs beyond an acceptable threshold will prove fatal for the cell as they will lead to cell apoptosis or accumulation of mutations with higher likelihood for cancer development [5, 21-24].” What do you mean by “beyond an acceptable threshold”? One unrepaired double strand break is lethal in most instances. What you probably mean is that too many DSBs overwhelm the DDR and will kill the cells because some will not be repaired. Please rephrase.

The reviewer is right. The phrase “beyond an acceptable threshold” is misleading and should be omitted. The correct phrase is: “Accumulation of DSBs will prove fatal for the cell as they will lead to cell apoptosis or accumulation of mutations with higher likelihood for cancer development”. We have now made the corresponding changes in the revised manuscript page 4.

  1. Why are BRCA1 and BRCA2 italicized but PARP1 is not?

Italics is used to describe genes. In this study, we evaluated BRCA1 gene expression and PARP1 protein expression.

  1. Please define CK in both introduction and abstract. Also describe this gene in the introduction.

Cytokeratin (CK) has now been defined in the abstract (page 1). It has also been defined in the introduction with the addition of the sentence “The epithelial marker, cytokeratin (CK) and particularly A45-B/B3 antibody (CK8, CK18, CK19) has been widely used for the characterization of cells as CTCs.” (Page 4).

  1. Please label Y axes for Figs 1, 2, 4, 5 and 6 with “Percent”.

According to the reviewer’s comment, we have now added Y axes to every histogram (pages 7, 9, 11, and Sup. Data).

  • Some minor typos have been identified and highlighted in pages 1, 3 and 7

Sincerely,

Galatea Kallergi

Round 2

Reviewer 2 Report

The authors have made significant changes to this version and this reviewer is satisfied. I have only a couple of minor points.

  1. For the graph figures, please see if you can move the numbers so that they don's overlay with the error bars. It looks a little odd.
  2. For figure 2, it might be best if you write the labels on each of the column of panels (e.g. DAPI, CK, etc).
  3. Also, for this second review, I received the supplemental tables (thank you for that) but the file does contain the supplemental figures. Please provide those as well.

Author Response

Reviewer’s comments

  1. For the graph figures, please see if you can move the numbers so that
    they don's overlay with the error bars. It looks a little odd.

The numbers have now been moved above the error bars in all the graphs.

  1. For figure 2, it might be best if you write the labels on each of the
    column of panels (e.g. DAPI, CK, etc).

Labels have now been placed on each of the column on each panel.

  1. Also, for this second review, I received the supplemental tables (thank
    you for that) but the file does contain the supplemental figures.
    Please
    provide those as well.

Supplementary Tables and Figures are now uploaded as one pdf.

Sincerely yours,

Galatea Kallergi

Assistant Professor of Biochemistry,

Department of Biology,

University of Patras, Patras, Greece
